# Temporal and Spatial Distribution and Fluorescence Spectra of Dissolved Organic Matter in Plateau Lakes: A Case Study of Qinghai Lake

**Zheng Li** [1,2,†], **Zhenghui Fu** [1,†], **Yang Zhang** [2], **Yunyan Guo** [1], **Feifei Che** [1], **Huaicheng Guo** [2,*] **and Shuhang Wang** [1,*]

1 National Engineering Laboratory of Lake Water Pollution Control and Ecological Restoration Technology, Chinese Research Academy of Environmental Sciences, Beijing 100012, China; 2001111921@stu.pku.edu.cn (Z.L.); fzh@pku.edu.cn (Z.F.); gyy@126.com (Y.G.); chefeifei@126.com (F.C.)

2 College of Environmental Sciences and Engineering, Peking University, Beijing 100871, China; zhangyang2019@pku.edu.cn

\* Correspondence: hcguo@pku.edu.cn (H.G.); wangshuhang@126.com (S.W.)

† Zheng Li and Zhenghui Fu made the same contribution to the article.

**Abstract:** Dissolved organic matter (DOM) has a great impact on the main pollution indicators of lakes (such as chemical oxygen demand, COD). Therefore, DOM is the research basis for understanding the meaning of the water environment and the laws of the migration and transformation of pollutants. Qinghai Lake is one of the world's typical inland plateau lake wetlands. It plays important roles in improving and regulating the climate and in promoting a virtuous regional ecological cycle. In recent years, with the acceleration of urbanization and the rapid development of tourism, under the background of climate change, and with grassland degradation and precipitation change, the whole basin of Qinghai Lake has been facing great ecological pressure. In order to comprehensively explore the water environment of Qinghai Lake and to protect the sustainable development of the basin, a systematic study was carried out on the whole basin of Qinghai Lake. The results show the following: (1) from 2010 to 2020, the annual average value of $COD_{Cr}$ in Qinghai Lake fluctuated in the range from class III to class V according to the surface water environmental quality standard, showing first a downward trend and then an upward trend. (2) The concentration of CDOM in Qinghai Lake had obvious temporal and spatial changes. (3) The spatial distribution of the total fluorescence intensity of FDOM in water was also different in different seasons. However, in the three surveys, the area with the highest total fluorescence intensity of FDOM in the water body appeared near Erlangjian in the south of Qinghai Province, indicating that anthropogenic sources are the main controlling factors of dissolved organic matter in the lake.

**Keywords:** dissolved organic matter; Qinghai Lake; chemical oxygen demand; colored dissolved organic matter; temporal and spatial distribution

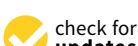



## 1. Introduction

Dissolved organic matter (DOM) is ubiquitous in soil, rivers, lakes, and oceans in nature. It is an important organic component in water (Figure 1) [1]. DOM is a complex organic mixture composed of different functional groups and molecular structures [2], which usually include humus, carbohydrates, and proteins, and it plays an important role in the global carbon cycle [3]. In natural water bodies, DOM may be derived from the decomposition of plant substances and extracellular substances released by aquatic organisms and microorganisms or from the transportation of degraded organic substances from the surrounding terrestrial environment by rivers and groundwater [4,5].

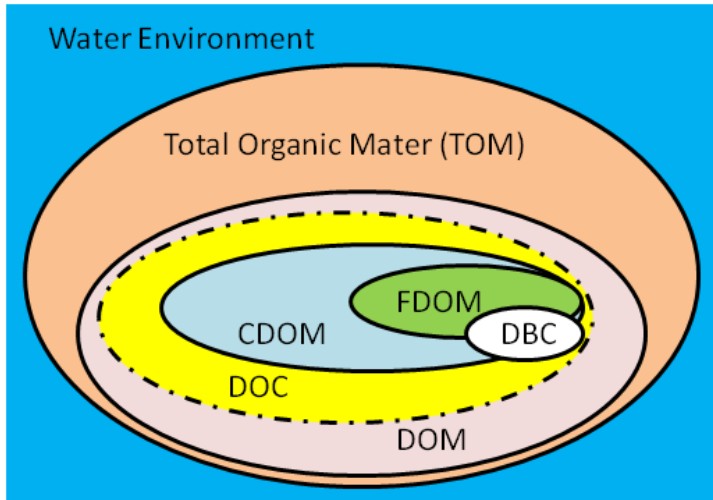

**Figure 1.** Schematic diagram of the main organic components in natural water (DOC is dissolved organic carbon; CDOM is colored dissolved organic matter; FDOM is fluorescent dissolved organic matter; DBC is dissolved black carbon).

CDOM is the main light-absorbing substance in dissolved organic matter (DOM) in natural water, and it can affect the migration and transformation of heavy metals and organic pollutants [6]. The part with fluorescence characteristics in CDOM is called fluorescent dissolved organic matter (FDOM), and it is of great significance for aquatic ecosystems [7]. As an important part of DOM, its important role in natural water environments has long been a concern of researchers. Foreign studies on CDOM were carried out before, and the research fields include the origin, destination, distribution, migration, and transformation of CDOM, as well as its spatio-temporal changes. At present, ultraviolet, visible, and fluorescence spectra are mainly used to study CDOM from different sources in water bodies at home and abroad. Moreover, the research on DOM is concentrated in inland basins, such as the Yellow River [8], the Yangtze River Estuary [9], and Songhua Lake [10], as well as reservoirs, thus focusing on unilateral research on the distribution characteristics of molecular weight, fluorescence characteristics, and correlations with metal ions. There is little research on typical ecological lakes on plateaus. This work is a systematic study of DOM in typical saline lakes on a plateau, and it provides supportive sediment data for the analysis of lake water environments and assessment of water bodies.

The Tibet Plateau is the highest plateau on Earth and has always enjoyed a reputation as the "Asian water tower" and "third pole". Its high terrain and unique geographical location have a significant impact on the global climate [11,12]. Qinghai Lake is an important water body for the maintenance of the ecological security of the Tibet Plateau. It plays a great role in improving and regulating climate and promoting a virtuous cycle of regional ecology [13]. However, there are few investigations or regional environmental studies on the water environment of the Qinghai Lake Basin. In recent years, with the acceleration of urbanization and the rapid development of tourism, Qinghai Lake Basin is facing great ecological pressure. Therefore, the investigation of the water quality of Qinghai Lake and the study of the temporal and spatial distribution and optical characteristics of dissolved organic matter in the water body are helpful in gaining a deep understanding of the current situation of the water environment of Qinghai Lake, the carbon cycle process within the basin, and its potential impact on water quality, and it can provide support in the form of background data for the analysis of lake water environments and assessments of water bodies.

## 2. Methodology

### 2.1. Study Area

Qinghai Lake is located in the northeast of the Qinghai Tibet Plateau, at the junction of Gangcha County, Gonghe County, and Haiyan County. It is the largest inland saltwater lake in China. In 2019, the lake's area reached 4519 km². The lake water is weakly alkaline, has a low oxygen content, has a salt content of 14.1 G·L$^{-1}$, and has transparency below 3 M. Qinghai Lake Basin is between 97°50′~101°20′ E and 36°15′~38°20′ N, with a drainage area of about 29,661 km² [14]. The overall outline is that of an oval, and the terrain is high in the northwest and low in the northeast. It is located at the intersection of the East Asian monsoon region, the northwest arid region, and the alpine region of the Qinghai Tibet Plateau. It belongs to the semi-arid and alpine climatic region of the plateau, and the precipitation is unevenly distributed throughout the year [14]. Qinghai Lake is located in the southeast of the basin and is mainly supplied by rivers and precipitation. Most of the rivers entering the lake are seasonal rivers and are concentrated in the west and north of the lake. The main supply rivers are Buha River, Shaliu River, Quanji River, Hargai River, Ganzi River, and HeMa River.

This study takes Qinghai Lake Basin as the research object, including 8 rivers entering the lake and the body of Qinghai Lake. The water quality data of the 8 rivers entering the lake were collected from 2010 to 2020, and the river distribution map is shown in Figure 2. There were 14 sampling points in Qinghai Lake, and the layout of the sampling points is shown in Figure 3. In May, September, and October 2020, Qinghai Lake Basin was comprehensively sampled three times; 2.5 L surface water samples (0–0.25 m from the water surface) were collected, stored in brown glass bottles (constant 4 °C), and sent back to the laboratory for testing.

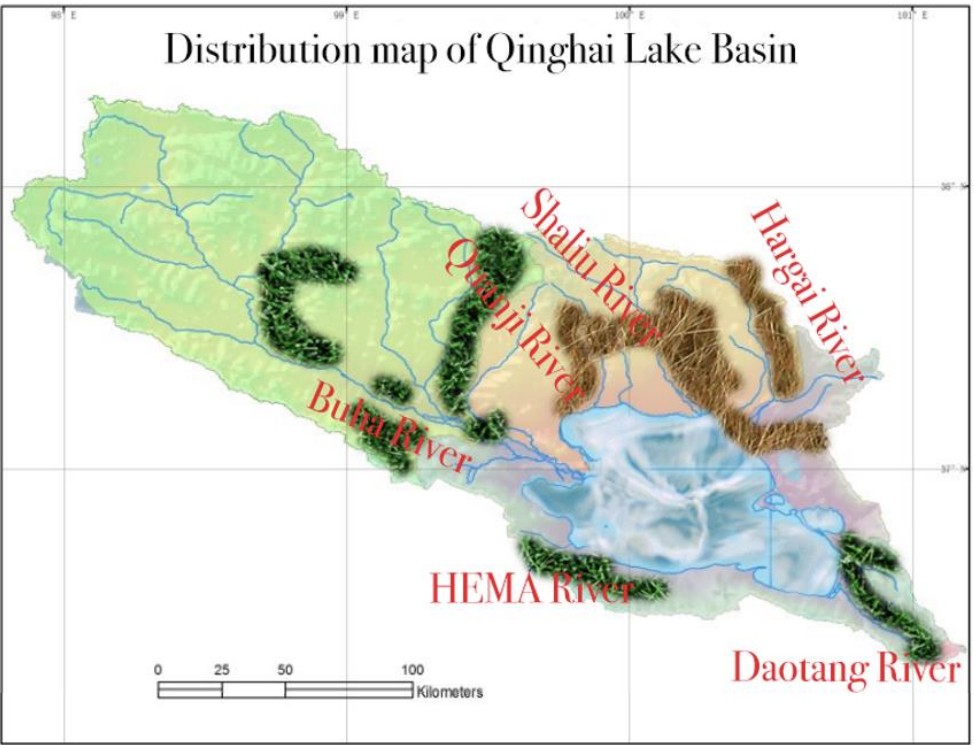

**Figure 2.** Distribution map of Qinghai Lake Basin.

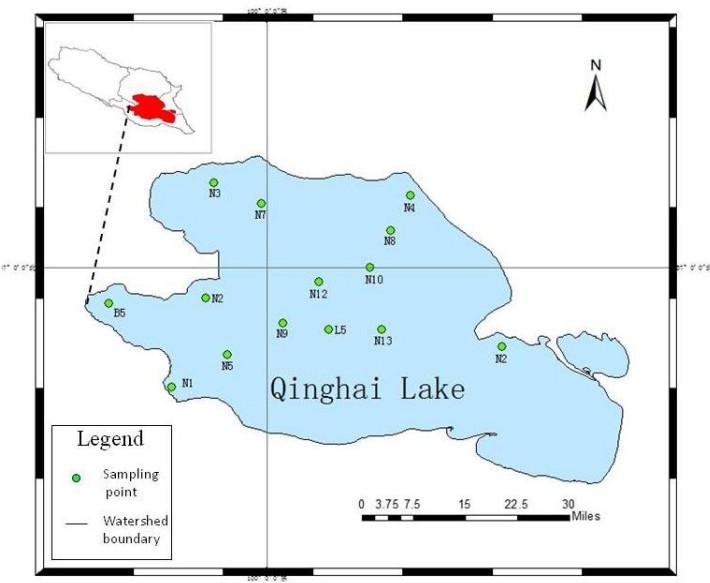

**Figure 3.** Locations of the DOM sampling points in Qinghai Lake.

### 2.2. Sampling Method

Seawater samples were collected with a Niskin water collector, and the parameters, such as the water depth, temperature, and salinity, were synchronously measured with a CTD sensor during the seawater collection. The river water samples were taken manually, and the parameters, such as the temperature, salinity, and pH value, of the on-site river water were synchronously measured in the river with a portable water quality analyzer. For the DOM and CDOM, filtering devices were used to filter the water samples to collect them in the shortest possible amount of time, and the filtered samples could be frozen and stored. They were thawed in the laboratory before measurement, and sufficient samples were transferred to a sample bottle for DOC measurements (24 or 40 mL) with a burned Pasteur glass pipette (Corning). If the sample could not be measured immediately, acid was added to the sample (pH < 2), and it was stored in cold storage for measurement.

### 2.3. Sample Analysis

In order to facilitate the examination of the instruments used in the experiment and the experimental process, the instruments used for the determination of the parameters can be found in Table 1, and the process is detailed in the following.

**Table 1.** Instruments used for the determination of the parameters.

| Measurement Index | Instrument/Method | Model | Origin/Manufacturer |
|---|---|---|---|
| Water temperature (T), pH, and dissolved oxygen (DO) | Multi-parameter water quality monitor | YSI 6600V2 | USA |
| potassium bichromate ($COD_{Cr}$) and permanganate ($COD_{Mn}$) | Titrimetric method | (MEP of PRC 1989) | China |
| TOC (DOM) | Total organic carbon analyzer | TOC-V | Shimadzu, Kyoto, Japan |
| CDOM | (UV-Vis) Ultraviolet–visible spectrophotometer | UV-2600PC | Shimadzu, Kyoto, Japan |
| FDOM | Fluorescence spectrum | Hitachi F-7000 | Hitachi, Tokyo, Japan |

Before the analysis of the DOM and CDOM, the water samples were filtered using pre-combusted (450 °C for 5 h) glass fiber filters (0.7 µm, GF/F, Whatman).

The absorption coefficient of CDOM at 254 nm ($\alpha 254$, m$^{-1}$) was used as a proxy for the CDOM concentration. The value of $\alpha 254$ was calculated with the following equation [15]:

$\alpha254$ 1/4 2:303A254 = L ð1Þ, where A254 is the absorbance at 254 nm and L is the path length of the cuvette (0.1 m). Fluorescence EEMs–PARAFAC analysis of fluorescent DOM (FDOM) was carried out with a 150 W Xe lamp as the excitation light source and a 9 PMT voltage of 700 V. The scanning ranges of the excitation wavelength (Ex) and emission wavelength (Em) were from 200 to 450 nm and 250 to 600 nm, respectively. The increments of the excitation and emission wavelength were all set to 2 nm. The slit width and the scanning speed were set to 10 nm and 12,000 nm·min$^{-1}$, respectively. In order to reduce the influence of instrument conditions and Raman scattering on the fluorescence spectra, an ultrapure water blank was used as a reference. The reagents used in the determination experiment were of excellent grade and pure. During the experiment, the instrument was calibrated with national first-class reference materials, and the standard curve was established to reach 99.99%. The standard solution was used for quality control, and parallel double-sample synchronous analyses were carried out. Each element of the analysis method met the requirements of the detection limit (CL), the confidence reached 95%, the relative standard deviation (RSD) was less than 5%, and the accuracy was good.

The analysis and test results were analyzed with Excel 2019, SPSS 24.0, and MATLAB 2020. The correlation diagram was completed with ArcGIS 10.2 and MATLAB 2020. Multivariate statistics and other methods were used to analyze and calculate the eigenvector to determine the components of different DOMs.

Firstly, the matrix spectrum data, which were characterized by three-dimensional fluorescence spectra, were preprocessed with the method of Wang et al. (2018), and then the preprocessed data were analyzed through parallel factor analysis (PARAFAC) of the three-dimensional fluorescence data with the MATLAB sub-software Solo. PARAFAC modeling was performed using MATLAB (MathWorks, Natick, MA, USA) with the DOM Fluor toolbox. The number of components was determined based on a core consistency test and split-half validation. The fluorescence index (FI) was the fluorescence intensity ratio of the fluorescence emission spectrum at 450 and 500 nm with an excitation light wavelength of 370 nm [16,17].

## 3. Results and Discussion

### 3.1. Time Variation of COD$_{Cr}$ and Permanganate Index in Qinghai Lake from 2010 to 2020

In order to clarify the background values of the Qinghai Lake watershed, indexes for monitoring water quality were collected and analyzed to determine the characterizations of the variations in water quality. According to the historical monitoring data on the water quality (Figure 4), the COD$_{Cr}$ concentration in Qinghai Lake was between 18.11 and 38.00 mg/L from 2010 to 2020, with an average of 27.10 mg/L, which was at the surface class IV level. Except for the large fluctuation of the COD$_{Cr}$ concentration in 2010 and 2011, the fluctuation of the COD$_{Cr}$ concentration in the water of Qinghai Lake in other years was relatively small. In terms of the mean values of the COD$_{Cr}$ at each historical monitoring point, the mean value of COD$_{Cr}$ at the Sha Dao point was the highest relative to the others with 43.84 mg/L.

From 2010 to 2020, the annual mean value of COD$_{Cr}$ of the water body of Qinghai Lake first decreased and then increased; it fluctuated greatly between the class III and V levels of surface water. Among them, the average value of COD$_{Cr}$ in 2011 was relatively the highest and was close to the upper limit of the class V water quality standard for surface water, and the annual average value of COD$_{Cr}$ in 2012 was relatively the lowest. In 2020, the COD$_{Cr}$ rose to the class IV water level, with an average of 36.02 mg/L.

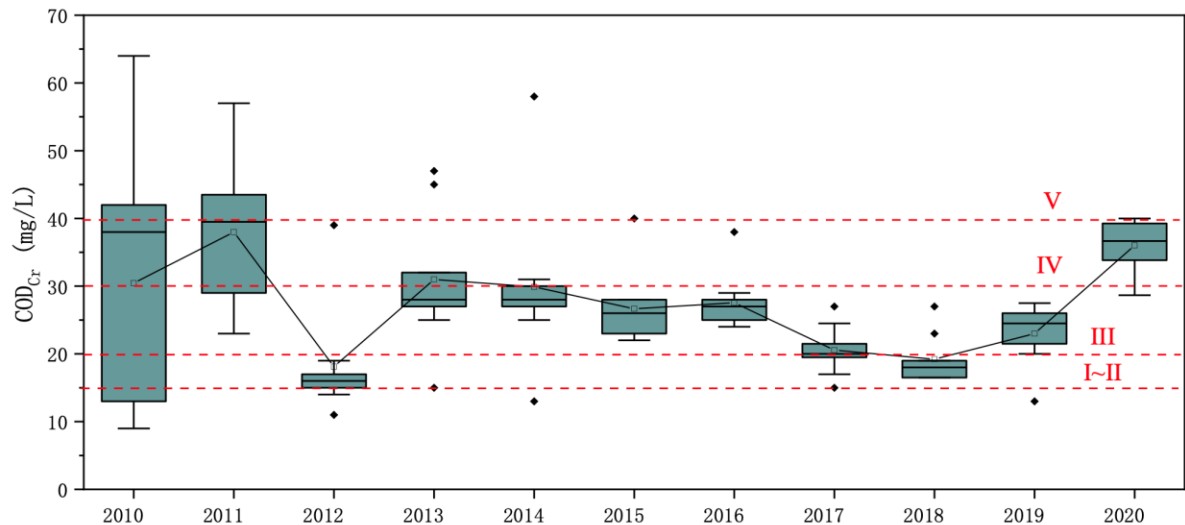

**Figure 4.** Change trend of $COD_{Cr}$ in Qinghai Lake from 2010 to 2020.

The on-site monitoring in May, September, and October 2020 showed that (Figure 5) the annual $COD_{Cr}$ of the water body of Qinghai Lake ranged from 31 to 48 mg/L, with an average of 39.21 mg/L. The $COD_{Cr}$ of all points in the water body exceeded the class IV standard for surface water (threshold ≤ 30 mg/L), and the $COD_{Cr}$ of at some points in the water body exceeded the class V standard for surface water (threshold ≤ 40 mg/L). From the perspective of monthly variation, the ranges of $COD_{Cr}$ in the water body of Qinghai Lake were 31–45, 32–47, and 31–48 mg/L, respectively, and the average values were 40.13, 39.43, and 38.09 mg/L. On the whole, the difference in $COD_{Cr}$ in the water body of Qinghai Lake in different months was small, and the $COD_{Cr}$ concentration was at a high level. It can be seen from Figure 4 that from 2018 to 2020, the water quality of the rivers entering Qinghai Lake gradually worsened from class II surface water in 2018 to class IV surface water in 2020. Combined with the detection data from the lake samples in 2020, it was found that the COD content in Qinghai Lake was similar to that in the rivers entering the lake. Though it is the largest saltwater lake in Western China, Qinghai Lake has a very small population, but there are many herdsmen living around the rivers entering the lake. Therefore, the main source of $COD_{Cr}$ in Qinghai Lake may be affected by the rivers entering the lake, non-point sources in the grasslands, and human activities in the surroundings.

According to the historical monitoring data (Figure 6), the permanganate concentration index in the water of Qinghai Lake from 2010 to 2020 was between 2.14 and 5.53 mg/L, with an average of 3.89 mg/L, which is at the class II level for surface water. Except for the large fluctuation of the permanganate concentration index in the water body in 2012, its fluctuation in the Qinghai Lake's water body in other years was relatively small. Among the mean values of the permanganate index at each historical monitoring point, the mean value at the Shadao point was relatively the highest at 5.39 mg/L.

From 2010 to 2015, the annual mean value of the permanganate index in the water body of Qinghai Lake fluctuated as a whole, but all values met the class III level for surface water. From 2016 to 2020, the annual average values of the permanganate index in the water body of Qinghai Lake showed a downward trend and was basically at the class II level for surface water. The permanganate index remained at the level of 2.14–2.35 mg/L from 2018 to 2020, which was slightly higher than the limit for class I water quality. The average value of the permanganate index in 2011 was relatively the highest at 5.53 mg/L, which was close to the upper limit of the class III water quality standard for surface water, and the annual average value of the permanganate index in 2019 was relatively the lowest.

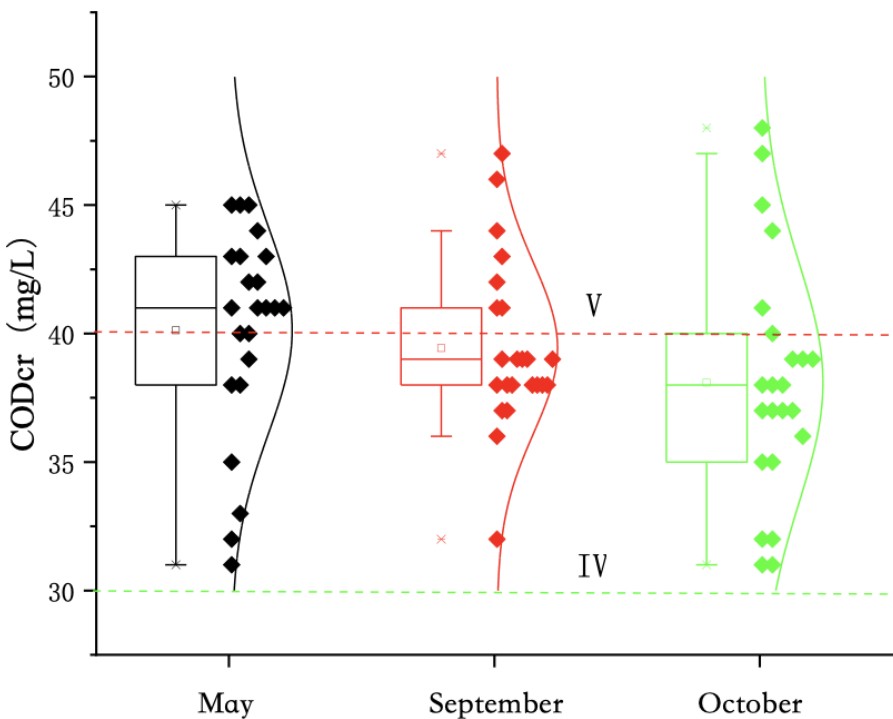

**Figure 5.** Characteristics of the COD$_{Cr}$ values in the water body of Qinghai Lake in different seasons.

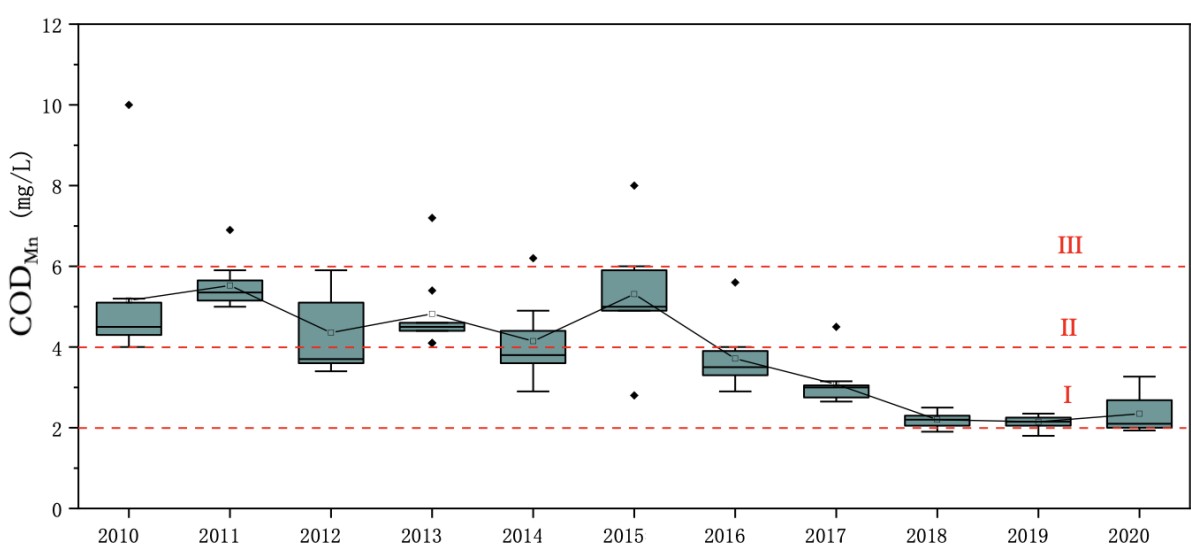

**Figure 6.** Change trend of the permanganate index in Qinghai Lake from 2010 to 2020.

*3.2. Time Variation of CODcr and Permanganate Index in the Rivers Entering Qinghai Lake from 2010 to 2020*

From 2010 to 2020, the COD$_{Cr}$ of the main rivers entering Qinghai Lake Basin was 7.50–13.94 mg/L, with an average of 10.99 mg/L. The COD$_{Cr}$ of the rivers entering the lake was lower than that of the lake as a whole (Figure 7).

Except for the DaoTang River, the mean values of the COD$_{Cr}$ of the major rivers were relatively similar from 2010 to 2020, and the interannual variation trend was relatively consistent. The eight main rivers entering Qinghai Lake from 2010 to 2020 were ranked in the following order according to their average CODcr: Ganzi River < Buha River < Jilmeng River < Hargai River < Shaliu River < Quanji River < HeMa River < DaoTang River.

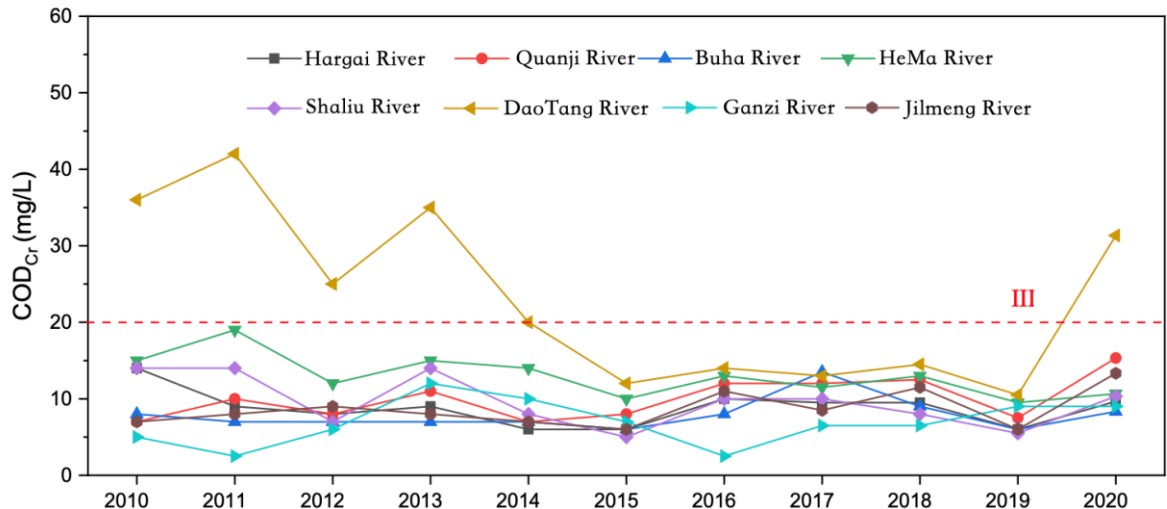

**Figure 7.** Change trend of COD$_{Cr}$ in the water bodies of rivers entering the lake from 2010 to 2020.

From 2010 to 2020, the permanganate index of the main rivers entering the lake in Qinghai Lake Basin was between 1.44 and 2.98 mg/L, with an average of 2.18 mg/L (Figure 8); The permanganate index of the rivers entering the lake was lower than that of the lake as a whole.

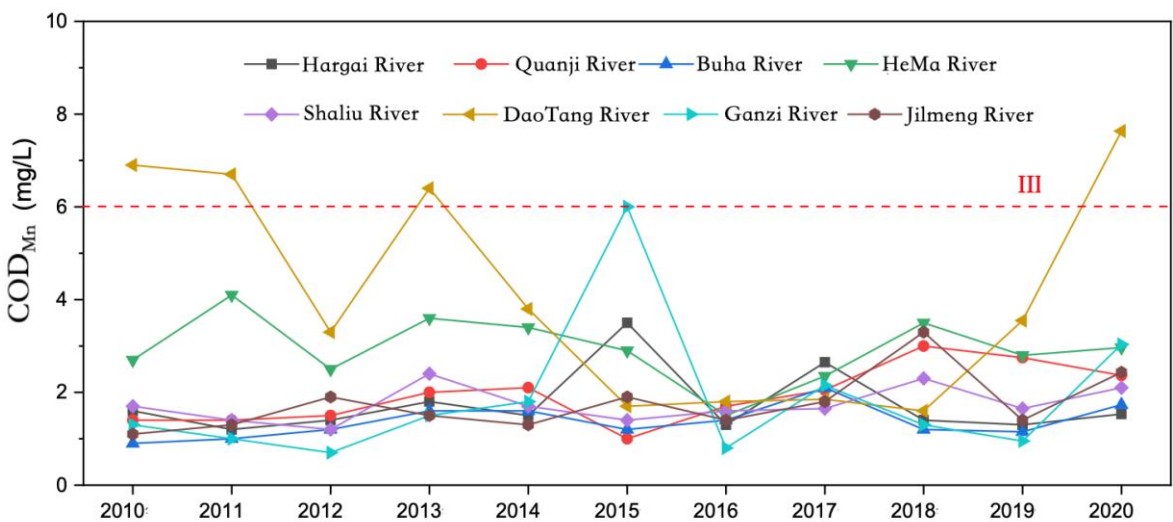

**Figure 8.** Change trend of the permanganate index of rivers entering the lake from 2010 to 2020.

Except for Daotang River and Heima River, the mean values of the permanganate index of the main rivers were relatively similar from 2010 to 2020, and the interannual variation trend was relatively consistent. The eight main rivers entering Qinghai Lake from 2010 to 2020 were ranked in the following order according to the average values of the permanganate index: Buha River < Shaliu River < Hargai River < Jilmeng River < Ganzi River < Quanji River < Heima River < DaoTang River. Except for Shaliu River and Ganzi River, the order of the other rivers was consistent with COD$_{Cr}$.

### 3.3. Spatial Distribution of COD$_{Cr}$ and the Permanganate Index in Qinghai Lake

There were obvious differences in the spatial distribution of COD$_{Cr}$ concentrations in Qinghai Lake (Figure 9). In May, the COD$_{Cr}$ was higher in the east than in the west, and that in the center of the lake was higher than that at the coast. In September and October, the COD$_{Cr}$ of the water body showed two high areas, namely, near the Jiangxi ditch wharf and Sand Island, as well as near the Jiangxi ditch wharf and Qinghai Lake fishing ground

wharf. From the perspective of the spatial distribution characteristics, the higher values in those three months were mainly concentrated at the Jiangxi ditch wharf and Qinghai Lake fishery wharf. The Jiangxi ditch wharf and Qinghai Lake fishery wharf are located in the south of Qinghai Province, and the planting industry is mainly along the south coast. Therefore, the input of external sources led to the higher concentration of CODCr in the water body in the south of Qinghai Lake than in other regions.

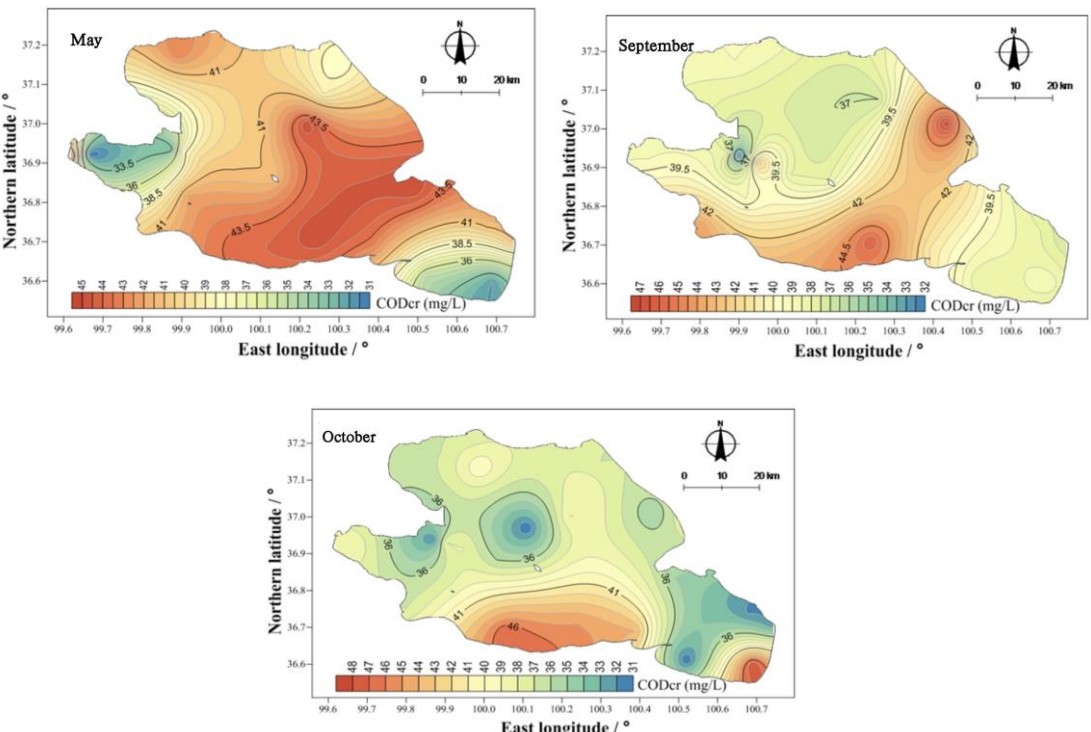

**Figure 9.** Spatial distribution characteristics of the $COD_{Cr}$ concentrations in Qinghai Lake in different seasons.

There were obvious differences in the spatial distribution of the permanganate concentration index in Qinghai Lake (Figure 10). In May, the range of the permanganate concentration index was 1.80–4.50 mg/L, with an average of 2.21 mg/L. In September, the range of the permanganate concentration index was 1.90–14.50 mg/L, with an average of 2.83 mg/L, and in October, the range of the permanganate concentration index was 1.80–4.50 mg/L, with an average of 2.29 mg/L. In May and September, the permanganate index in the east was higher than in the west. In October, the permanganate index of the water body showed three high areas, namely, the center of the lake, the east and the west, and the shore area close to the center of the lake. From the perspective of the spatial distribution characteristics, except for the center of the lake in October, areas with high values appeared close to the shore. Therefore, external inputs may be the reason for the higher concentrations of permanganate in the water body of Qinghai Lake than in other areas.

*3.4. Optical Properties of the DOM in Qinghai Lake*

Colored soluble organic matter (CDOM) and fluorescent soluble organic matter (FDOM) are two important indicators of the optical properties of DOM. CDOM is a component of DOM that can absorb effective ultraviolet and photosynthetic radiation; FDOM is a component of DOM that can release fluorescent photons after absorbing light radiation. Therefore, CDOM and FDOM are commonly used to characterize the optical characteristics of DOM.

(1)　Seasonal and spatial distributions of CDOM

The concentration of CDOM was characterized by its absorption coefficient, a254, at 254 nm. The field monitoring data in May, September, and October 2020 show that (Figure 11) the values of a254 in the water body of Qinghai Lake were between 8.80 and 13.29 M$^{-1}$, with obvious seasonal differences. Among them, the value of a254 in the water in May was between 8.80 and 9.83 M$^{-1}$, with an average value of 9.17 M$^{-1}$, which was significantly lower than in other seasons ($p < 0.01$). In September, the value of a254 in the water body was between 8.73 and 10.64 M$^{-1}$, with an average value of 9.48 M$^{-1}$. In October, the value of a254 in the water body was between 9.14 and 13.29 M$^{-1}$, with an average value of 9.81 M$^{-1}$, showing the ranking: October > September > May. The concentration of CDOM in Qinghai Lake not only had obvious seasonal differences, but also showed different spatial differences. The highest values of CDOM at each sampling point in the water body in May, September, and October appeared near Erlangjian in the south of the lake. In addition, the value of CDOM in the water body on the west bank of Qinghai Lake in May was also high.

(2)　Fluorescence spectra and fluorescence component characteristics of FDOM

The three-dimensional fluorescence spectrum matrix data of the DOM in Qinghai Lake were analyzed using the PARAFAC model, and four DOM fluorescence components with a single maximum emission wavelength were analyzed. The distribution of the maximum excitation/emission wavelengths of the four fluorescent components and the three-dimensional fluorescence spectra of the principal components are shown in the figure below (Figure 12).

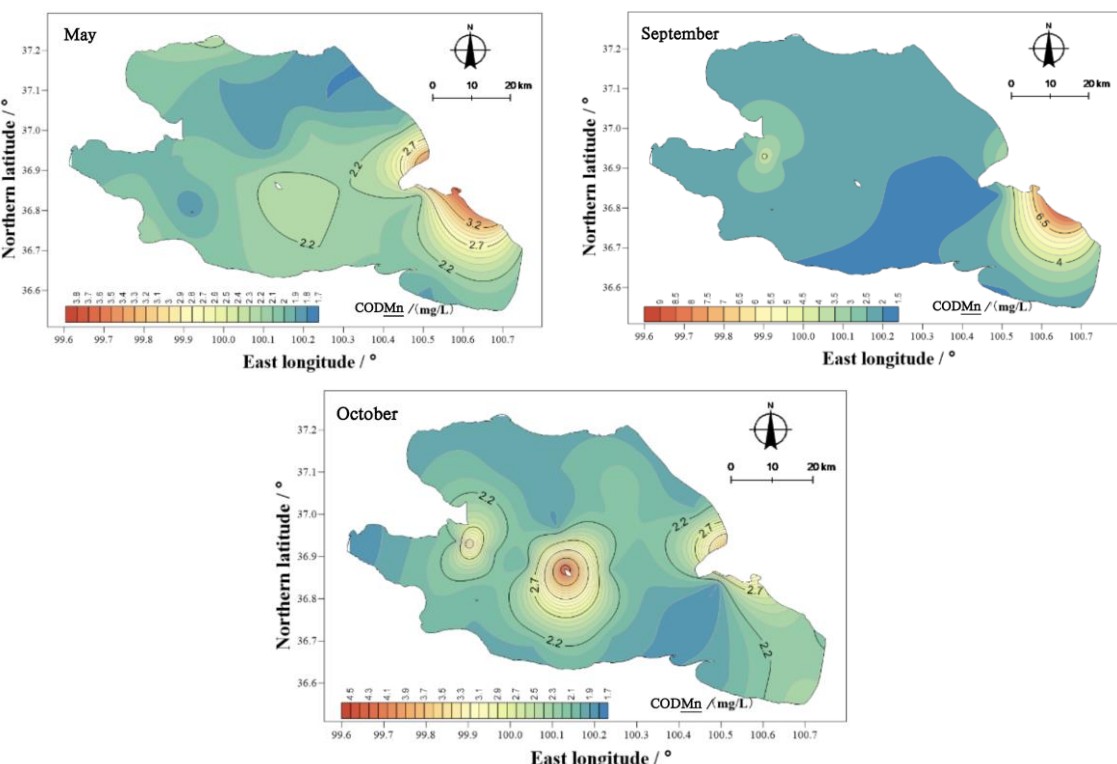

**Figure 10.** Spatial distribution characteristics of the permanganate concentration index in Qinghai Lake in different seasons.

The excitation wavelength of component C1 was at 242 nm, and the maximum emission wavelength was near 422 nm, which reflected the land-based high-molecular-weight humic acid, which was mainly from the degradation of higher plants or soil leaching. Component C2 had two obvious excitation wavelengths at 220 and 272 nm, and the maximum emission wavelength was 422 nm, which reflected the fluorescence peak of low-molecular-

weight fulvic acid, which mainly came from the fluorescence peak formed by biodegradable organic compounds. Component C3 had two obvious excitation wavelengths at 232 and 286 nm, and the maximum emission wavelength was 338 nm. It had binding peaks similar to those of protein and fulvic acid, which have a red shift compared with the conventional tryptophan-like peak. Component C4 had an obvious excitation wavelength at 270 nm and the maximum emission wavelength was 478 nm, reflecting the fluorescence peak formed by the polymer humic acid.

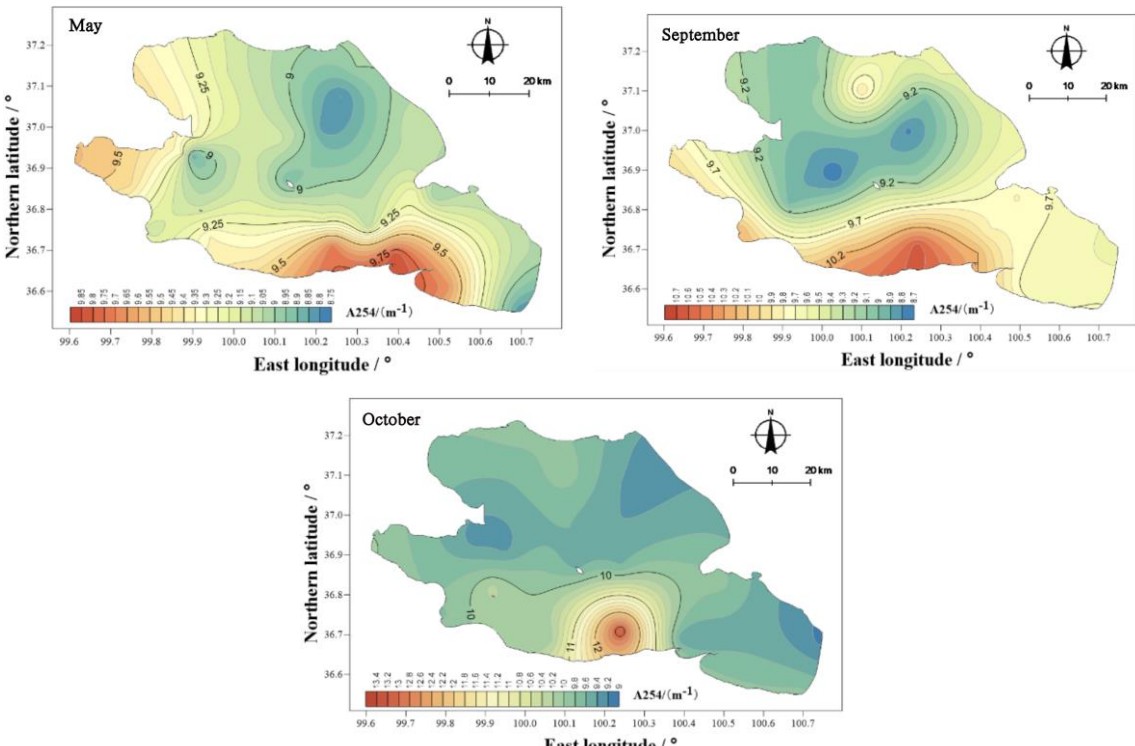

**Figure 11.** Spatial distribution characteristics of the CDOM concentration ($a_{254}$, m$^{-1}$) in the water body of Qinghai Lake in different seasons.

Overall, the proportion of each component in the total fluorescence intensity of FDOM in May and September was C2 > C1 > C4 > C3, and the proportion of each component in the total fluorescence intensity of FDOM in October was C3 > C2 > C1 > C4, which had certain seasonal differences (Figure 13).

In the three sampling surveys of Qinghai Lake in May, September, and October (Figure 14), the average total fluorescence intensity of FDOM in the water body of Qinghai Lake was 22.20, 24.17, and 42.80 R.U., showing significant seasonal differences: October > September > May. The spatial distribution of the total fluorescence intensity of FDOM in the water was also different in different seasons. The spatial difference in October was more obvious than those in May and September. However, in the three surveys, the highest value of the total fluorescence intensity of FDOM in the water appeared near Erlangjian in the south of Qinghai Province.

The mean values of component C1 in the overlying water of Qinghai Lake in May, September, and October were 5.85, 6.86, and 7.23 R.U. (Figure 15), showing the ranking of October > September > May. The spatial distribution characteristics were similar to those of the total amounts, and the spatial differences were obvious. The highest value appeared near Erlangjian in the south of Qinghai, and the lowest value appeared in the center of the lake.

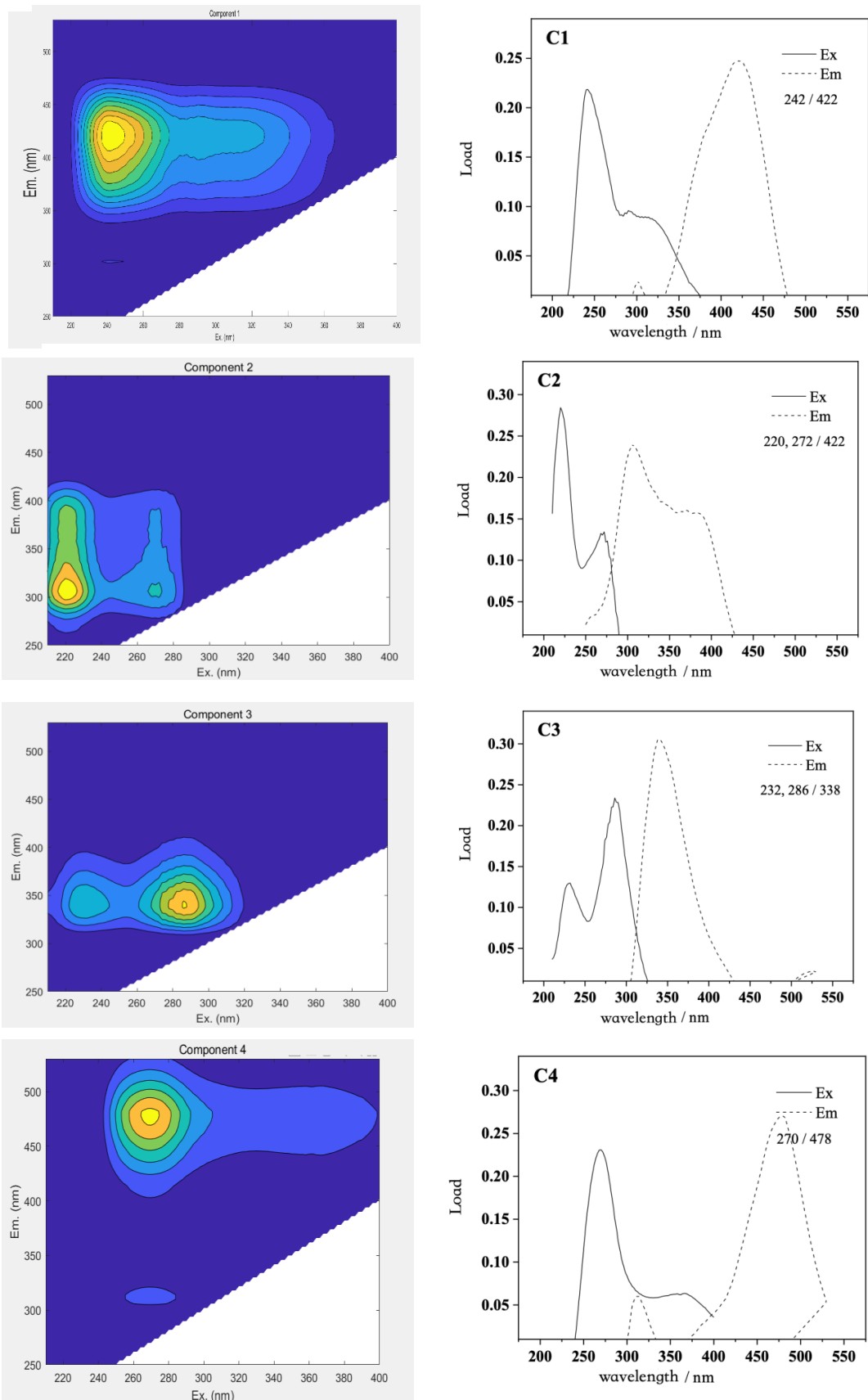

**Figure 12.** Characteristics of the fluorescent components of DOM in Qinghai Lake.

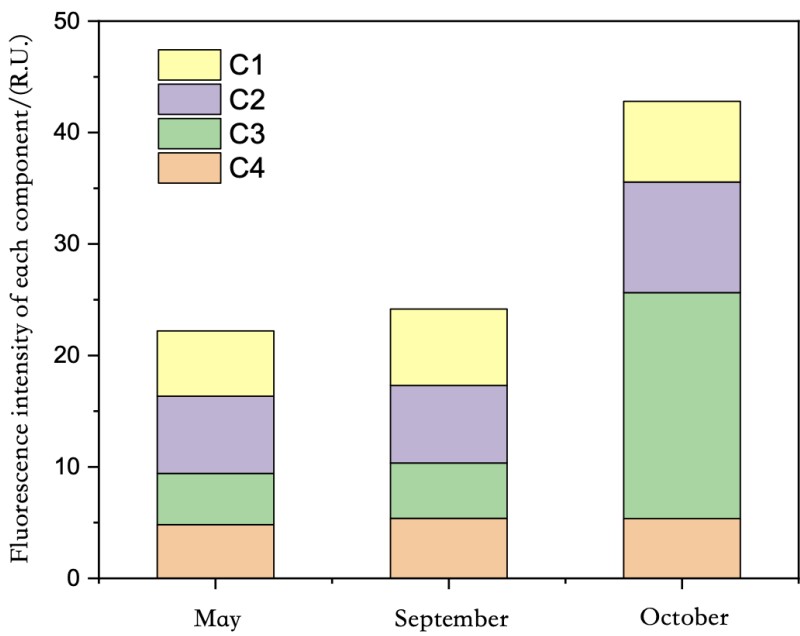

**Figure 13.** Fluorescence intensity characteristics of components of DOM in Qinghai Lake.

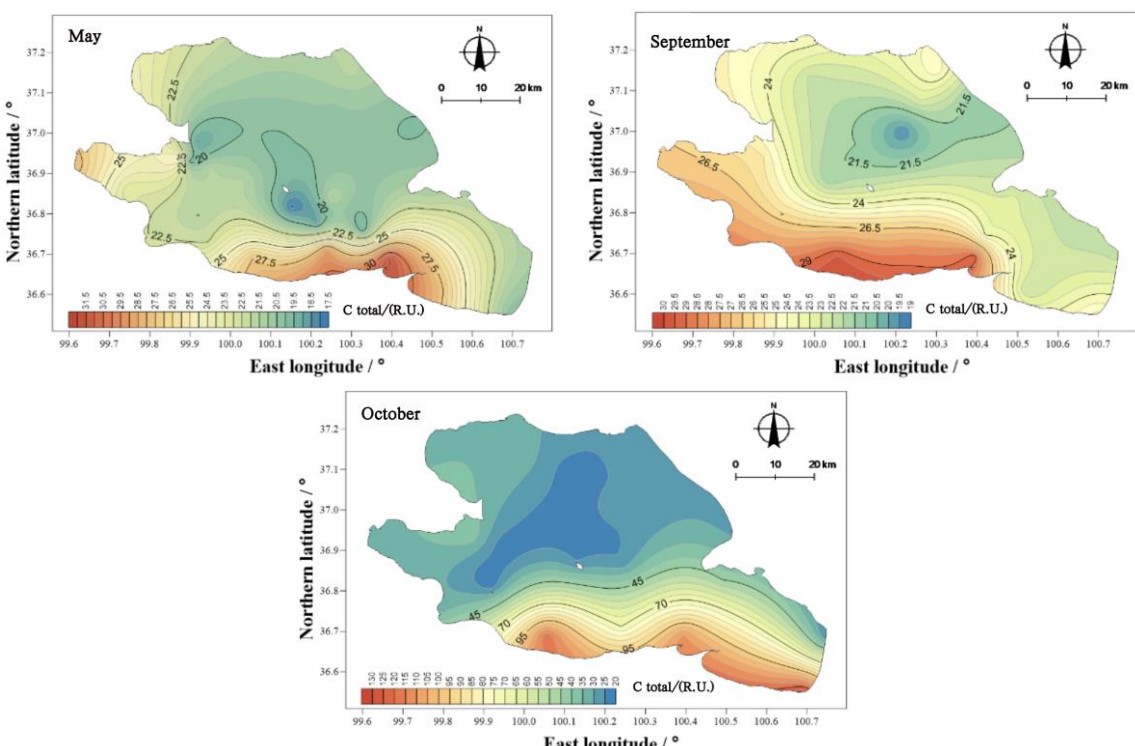

**Figure 14.** Spatial distribution of the total fluorescence intensity of FDOM in Qinghai Lake in different seasons.

The mean values of component C2 in the water body of Qinghai Lake in May, September, and October were 6.93, 6.96, and 9.93 R.U. (Figure 16), thus showing the ranking of October > September > May. The spatial distribution characteristics are similar to those of the total amounts, and the spatial differences were obvious. The highest values in May and October appeared in the south of Qinghai, and the highest values in September appeared in the southwest of the lake.

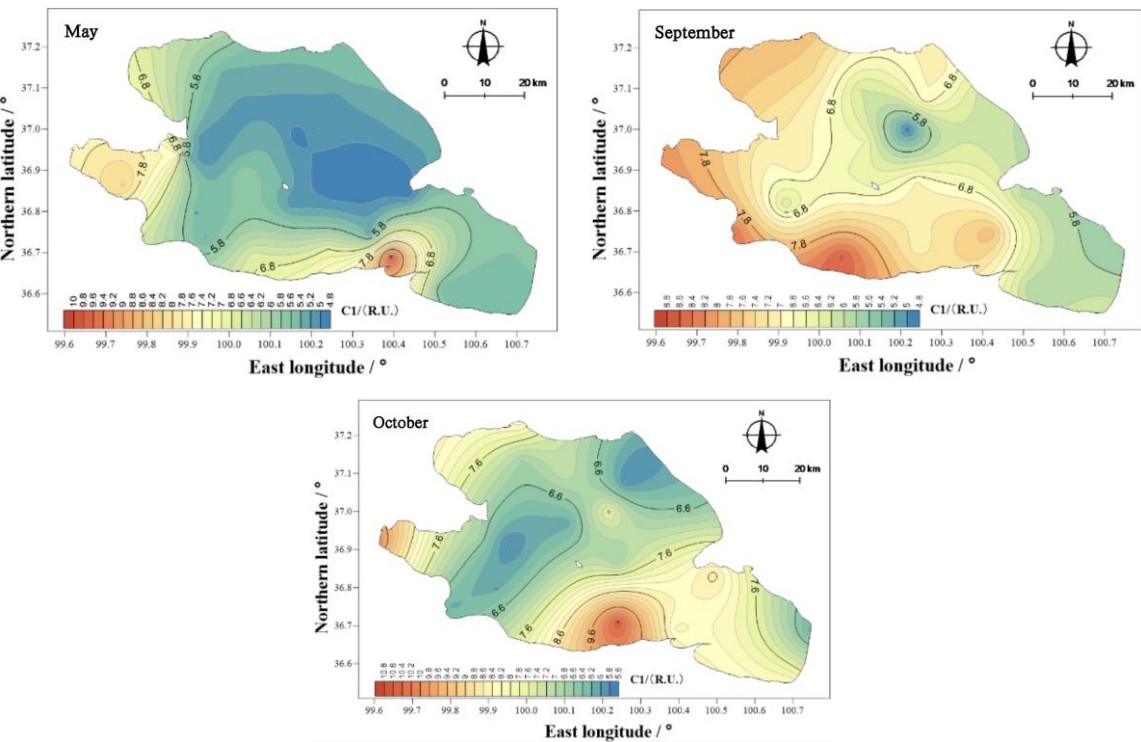

**Figure 15.** Spatial distribution characteristics of the fluorescence intensity of component C1 in Qinghai Lake in different seasons.

The mean values of component C3 in the water body of Qinghai Lake in May, September, and October were 4.59, 4.96, and 20.28 R.U. (Figure 17), thus showing the ranking of October > September > May. The spatial distribution characteristics were similar to those of the total amounts, and the spatial differences were obvious. The highest values appeared near Erlangjian in the south of Qinghai and Hunan.

The mean values of component C4 in the water body of Qinghai Lake in May, September, and October were 4.82, 5.39, and 5.36 R.U. (Figure 18), respectively, thus showing a different ranking from that of the other components, that is, September > October > May. The spatial distribution characteristics were similar to those of the total amounts, and the spatial differences were obvious. The highest values appeared near Erlangjian in the south of Qinghai and Hunan.

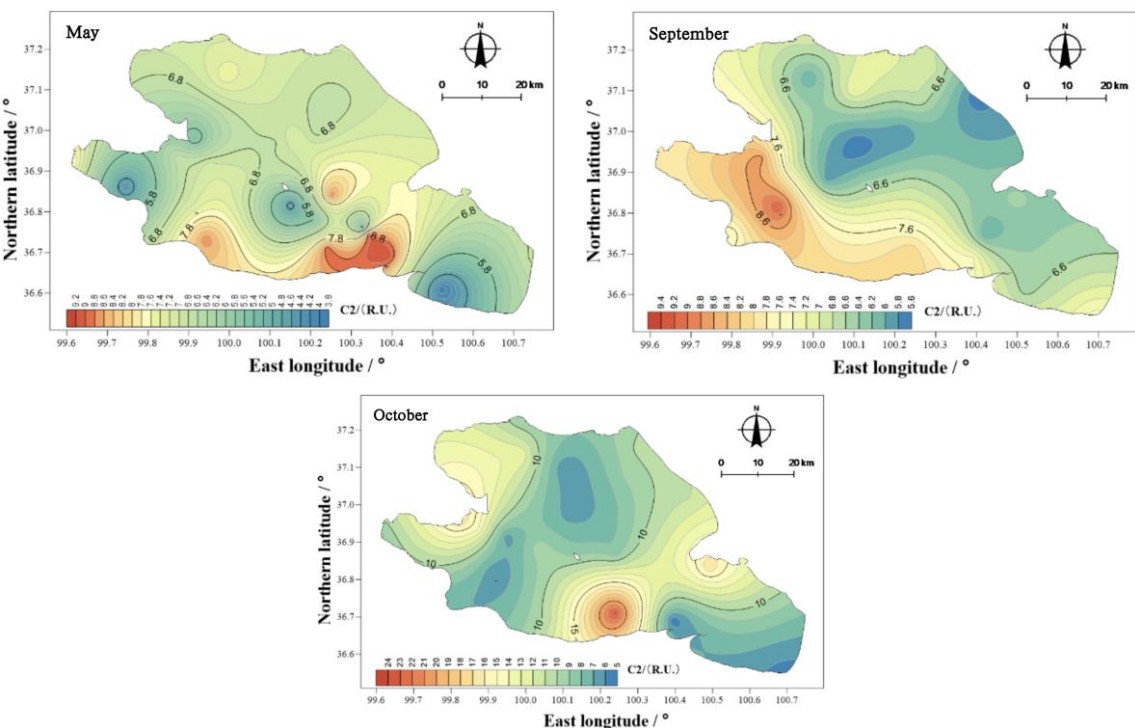

**Figure 16.** Spatial distribution characteristics of the fluorescence intensity of component C2 in Qinghai Lake in different seasons.

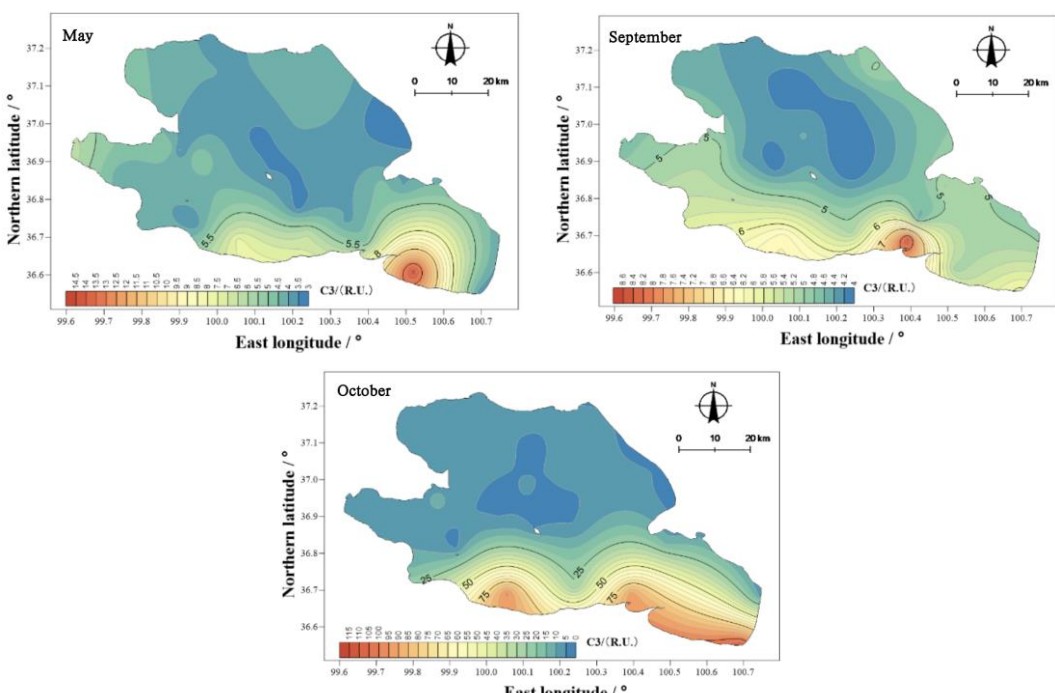

**Figure 17.** Spatial distribution characteristics of the fluorescence intensity of component C3 in Qinghai Lake in different seasons.

### 3.5. Bioavailability of DOM in Qinghai Lake

Both COD and $BOD_5$ are used to quantitatively reflect the degree of organic pollution in water. Previous studies have shown that when $BOD_5/COD_{Cr} \geq 30\%$, it is biodegradable sewage. If $BOD_5/COD_{Cr} < 30\%$, it is difficult to biodegrade the sewage. In this study (Figure 19), the $BOD_5/COD_{Cr}$ values in Qinghai Lake were between 1.90% and 5.29%, with an average of 3.36%, which was obviously low, indicating that the bioavail-

ability of organic matter in Qinghai Lake was poor and that it was difficult to decompose. This non-decomposable organic matter will continue to accumulate in the sediment over time and will migrate between the water and sediment due to disturbances from wind, waves, and fish, thus also reflecting the reason for the high COD of Qinghai Lake from another perspective.

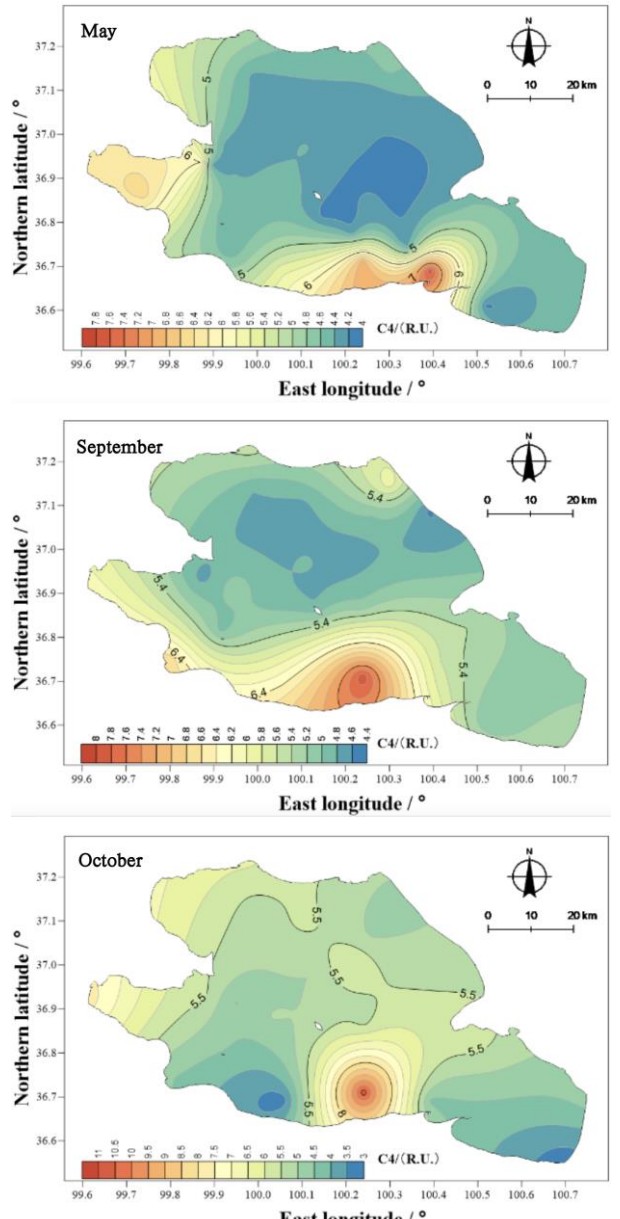

**Figure 18.** Spatial distribution characteristics of the fluorescence intensity of component C4 in Qinghai Lake in different seasons.

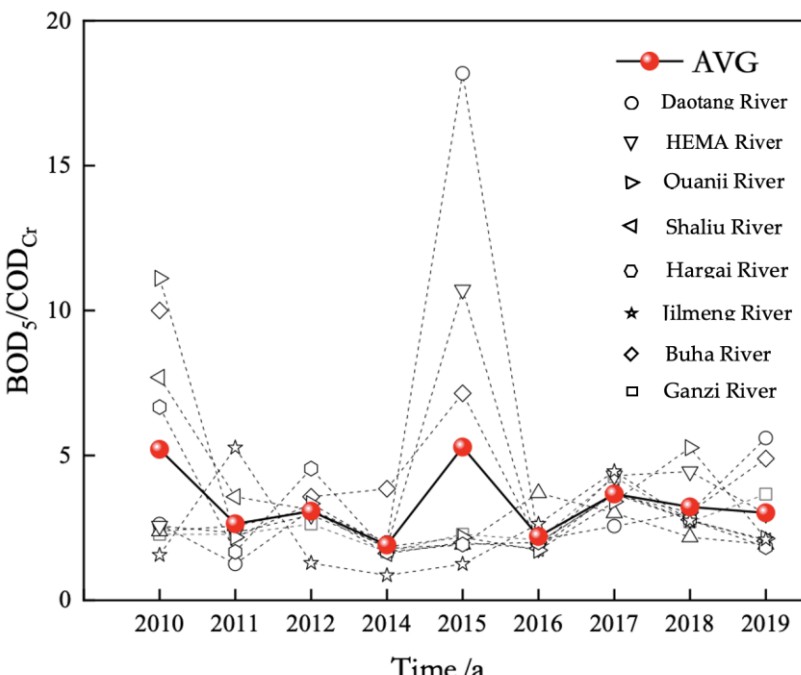

**Figure 19.** Bioavailability of DOM in Qinghai Lake.

## 4. Conclusions

(1) The high values of $COD_{Cr}$, $COD_{Mn}$, and BOD in Qinghai Lake are mainly due to the influence of the rivers entering the lake, non-point grassland sources, and human activities in the surroundings. (2) The order of the input of pollutants from eight rivers into Qinghai Lake was Ganzi River < Buha River < Jilmeng River < Hargai River < Shaliu River < Quanji River < HEMA River < Daotang River. Exogenous inputs are the main reasons for exceeding the standards for pollutants in Qinghai Lake. It is noteworthy that, as the main river entering Qinghai Lake, the treatment of Daotang River's water environment and ecological security becomes the main condition for sustainable development in the region. (3) The spatial distributions of the total fluorescence intensity of FDOM in the water were also different in different seasons. It is noteworthy that the area with the highest total fluorescence intensity of FDOM in the water body appeared near Erlangjian in the south of Qinghai Province, indicating that anthropogenic sources are the main controlling factors of dissolved organic matter in the lake. (4) By analyzing the three-dimensional fluorescence spectrum matrix data of the DOM in Qinghai Lake with the PARAFAC model, it was found that the DOM of Qinghai Lake was mainly composed of four material forms, namely, component C1, which reflects terrestrial high-molecular-weight humic acid, which is mainly from the degradation of higher plants or soil leaching; component C2, which has the fluorescence peak of low-molecular-weight fulvic acid that mainly comes from the fluorescence peaks formed by biodegradable organic compounds; component C3, which has protein-like and fulvic-acid-like binding peaks and has a red shift compared with the traditional tryptophan-like peak; and component C4, which reflects the fluorescence peak formed by the polymer humic acid.

**Author Contributions:** Z.L.: Conceptualization, Methodology, Writing, Software. Z.F.: Data curation, Writing—Original draft preparation. Y.Z.: Visualization, Investigation. Y.G.: Supervision. F.C.: Software, Validation. S.W.: Writing—Reviewing and Editing. H.G.: Writing—Reviewing and Editing. All authors have read and agreed to the published version of the manuscript.

**Funding:** Not Finding.

**Institutional Review Board Statement:** Not applicable.

**Informed Consent Statement:** Not applicable.

**Data Availability Statement:** Data available on request due to restrictions eg privacy or ethical. The data presented in this study are available on request from the corresponding author. The data are not publicly available due to: The original data of the manuscript is obtained through communication between the Chinese Academy of Sciences and local governments and on-site sampling. The data is very valuable and confidential. The data package is stored in the Chinese Academy of Environmental Sciences and local governments. Therefore, if the state or individual has legitimate reasons, they can contact the corresponding author.

**Conflicts of Interest:** The authors declare no conflict of interest.

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
