# Peer review of "Temporal and Spatial Distribution and Fluorescence Spectra of Dissolved Organic Matter in Plateau Lakes: A Case Study of Qinghai Lake"

_water, doi:10.3390/w13243481_

Round 1

Reviewer 1 Report

The paper reports on a lot of high-quality, systematic work. The Introduction, Methodology, and Results parts are strong and well-illustrated.

Unfortunately, the writing style is poor, and the valuable results are vaguely discussed or not discussed at all (repeating the results is not discussing them).

I recommend a thorough revision of the paper:

1) Improve the English (wording, capitalization, spacing, etc.). Explain abbreviations.

2) In the Abstract identify why the study was initiated, and what findings are new and original.

3) In the Introduction include findings from studies done in the literature, emphasizing the similarities and the differences.

4) The Results are solid and well-presented. However,  the Discussion section is weak: the new results should be compared to findings in other, similar studies, pointing out and explaining the similarities and differences.

5) The Conclusions has to repeat the most important new and original findings and their significance from theoretical and practical points of view.

In summary: this is valuable work that needs to be thoroughly revised in order to be properly presented in a scientific manner.

Reviewer 2 Report

The ms water-1475456 with the title of Temporal and spatial distribution and fluorescence spectra of dissolved organic matter in plateau lakes -- a case study of Qinghai Lake is well organized. The water quality monitoring data of Qinghai Lake water body and 8 rivers around the lake from 2010 to 2020 were collected, and the dissolved organic matter was synchronously sampled in May, September and October 2020. The optical characteristics of DOM, the temporal and spatial distribution of CDOM and the fluorescence spectrum and fluorescence component characteristics of FDOM were analyzed and studied.

First note for me was the type of the ms. The type of the ms should be Original article instead of Review? It is case study, but it is not review article! Am I right?

L3 remove one of these - -

What is COD? Authors have to write full words in the first mention, then abbreviations can be used.

In Keywords authors have to choose to include abbreviations of full words, and I recommend to keep only full words and remove the abbreviations.

The authors have to follow the guidelines and format of the journals in terms of citing the references within the ms. The journal use numerical format for the references.

L54-60 who said this? Please cite the relevant references

Change ml to mL within the whole ms.

The results section was well written and organized, as well as the Figures were very nice and clear. However, the discussion part was not presented and there were not other investigations interpreted with the current findings. Author must focus on the discussion section and make is deep.

For the conclusion, author should focus on the most important findings and make the conclusion shorter than the current form.

References are not written according to the format of the journal. In addition, the year was repeated twice in each reference. Authors have to revise this issue and pay attention the the writing process.

Authors should add the statistical analysis part at the end of material and methods section.

Good luck, Reviewer.

Reviewer 3 Report

Authors attempted to assess the temporal and spatial distribution and fluorescence spectra of dissolved organic matter in Qinghai Lake. It is a very simple study. Authors presented many results but not tried to justify their findings. Overall, it seems just dumping of results obtained through analysis of some sampling data. Besides, the paper is full of ambiguity. My major comments are given below.

  1. The methods used in the paper are not mentioned clearly. I must say it is not written completely. what is PARAFAC or EEMs-PARAFAC model? Nowhere mentioned in the paper. What does this do? It is just mentioned that PARAFAC modeling was performed using MATLAB. I couldn't even find the full meaning of PARAFAC in the whole paper.

  1. How have you prepared the spatial distribution maps from point data? What interpolation method was used? My major concern is that the estimated surface maps presented in the paper are wrong, whatever the method used. For example, no sampling data is available on the southeast side but showed low values of all parameters at the southeast corner. It seems values are considered low where no data is available. It is not the case in practice. Noway, any of the surface maps are acceptable.

  1. In Figures 4.1 and 4.2, the authors showed trends in DOM in water bodies of rivers entering the lake. Where are those rivers? It is not shown in the Lake map in Figure 2. Besides, are those points are at the mouth of the rivers? How you have estimated those values for different rivers. It is completely ambiguous.

  1. The paper is just a presentation of results; no analysis of results are there. Why such spatial and temporal variability in COD or other parameters? Why are some decreasing? Why variability of some of those is less in late period than the earlier period. I can't accept the results without any justification. It is a scientific paper, not just reporting results.

  1. Figure 3.2: Only in this figure the data points are shown. I counted there are 23 data points for each month. You mentioned you collected data for 11 years (2010 to 2020) at 14 sampling points in each of those months (May, September and October). Then why it is only 23 points? I believe data is not analyzed properly.

  1. How have you selected the sampling points? Those are well distributed over the lake. Why have you not collected any data from the south of the lake? The results do not present the dissolved organic matter in the lake. Besides, why have you selected those three months (May, September and October) for sampling?

  1. The authors collected many data. Good statistical analysis could be conducted. Interrelations among the parameters, how they are related to temperature, or with the season, etc. could be good information. But authors have not tried any critical analysis. I suggest authors do such analyses to make it a good paper.

  1. The paper is written in a very careless manner. There are many Chinese scripts within the figure. For example, Figures 4.3, 4.4, 5.1, 5.4, 5.5, 5.6, 5.7 and 5.8. This is not a Chinese journal. Figure numbering is also peculiar. I don’t think it is the standard numbering followed by the journal. Sometimes it is not matching with that mentioned in the text. For example, Figure 2-1 is cited in the text, but there is no such figure.
  2. Abstract is also written in a very novice manner. Author told almost nothing about the method. They mentioned results using four points, but the last point is not the result. They just mentioned the PARAFAC model used for analysis in point 4. I never saw any abstract finish with the method.

Though the paper is not sound, I believe such a study in a plateau lake is important. Therefore, I suggest the author improve the paper based on my comments and resubmit it.  

Round 2

Reviewer 1 Report

The authors properly and thoroughly revised their manuscript and responded to all my suggestions and comments in a highly professional manner.

In my opinion, the paper is publishable in its current, revised form.

Reviewer 2 Report

Although, the ms has been improved in terms of english, but still the authors did not follow the guidelines of the journal in terms of citations within the text.
